# Detection and density of breeding marsh birds in Iowa wetlands

Rachel A. Vanausdall ⓘ *◉, Stephen J. Dinsmore◉

Department of Natural Resource Ecology and Management, Iowa State University, Ames, IA, United States of America

◉ These authors contributed equally to this work.
* Rachel.Vanausdall@colostate.edu

## Abstract

Accounting for imperfect detection is an important process when obtaining estimates of density or abundance for breeding birds, and this is particularly true when researchers are monitoring birds to assess the success of restored wetlands. Due to the dramatic decline in areal cover and habitat quality, wetland restoration in the Prairie Pothole Region (PPR) is critically important to breeding birds. The Shallow Lakes Restoration Project (SLRP), a partnership between the Iowa Department of Natural Resources and Ducks Unlimited, Inc., aims to restore degraded shallow lakes throughout the Iowa PPR. We conducted unlimited-radius point counts with call-broadcast surveys for breeding marsh birds at 30 shallow lakes in various stages of restoration in 2016 and 2017. Our goals were to assess the impact of covariates on detection probability and estimate density of these species at non-restored, younger (1–5 years since restoration), and older (6–11 years since restoration) restorations. Detection probability ranged between 0.07 ± 0.009 (SE) for Red-winged Blackbird and 0.40 ± 0.09 (SE) for Common Yellowthroat. Percent cattail had a positive quadratic effect on detection probability for four species, with detection decreasing sharply as percent cattail increased and increasing slightly with 100% cattail cover. Wind speed negatively influenced the detection probability of Pied-billed Grebes but had a negative quadratic effect on the detection probability of Marsh Wrens. Both restored shallow lakes had greater densities of breeding Pied-billed Grebes, Marsh Wrens, and Yellow-headed Blackbirds than non-restored shallow lakes, but there was no significant difference between younger and older restorations. Including both habitat and environmental covariates on models for detection probability can improve the precision of estimates for density and should be considered when assessing bird populations pre- and post-restoration of shallow lakes.

**Data Availability Statement:** Excel files of the distance data for each species, along with the environmental and habitat covariates, are openly available through Iowa State University's DataShare (https://doi.org/10.25380/iastate. 11503833).

## Introduction

Effectively monitoring wildlife to assess the impact of management or restoration can be critical to identifying population trends or the need for further management action. Typically, monitoring involves surveys repeated over space or time to get estimates of abundance or density. An important component of most wildlife surveys and obtaining estimates of density is

**Funding:** Funding for this project was provided by the U.S. Fish and Wildlife Service (Prairie Pothole Joint Venture, grant number F15AP00612) and the Iowa Department of Natural Resources (Resource Enhancement and Protection License Plate Funds) to S. J. Dinsmore. The funders had no role in study design, data collection and analysis, decision to publish, or preparation of the manuscript.

**Competing interests:** The authors have declared that no competing interests exist.

the probability of detecting a species, as most surveys do not count every individual present [1,2]. Several sampling and analysis methods for estimating abundance or density when detection probability is less than one exist, and accounting for factors that influence detection probability can improve precisions of estimates [3–5]. This is particularly true for marsh birds, a group of birds that includes passerines and secretive marsh birds (i.e., rails, bitterns, grebes) and that can be difficult to monitor due to their elusive nature and the difficulty associated with surveying in wetland habitats [6,7], especially in large wetlands. Marsh birds are often considered wetland quality indicators [8] and some are experiencing declining populations [9,10], so obtaining reliable estimates of density can have important implications for wetland management and restoration.

Wetland restoration is a widely-used management tool for improving water quality and conditions for breeding marsh birds [11–13], and this is particularly true in the Prairie Pothole Region (PPR). Of interest throughout parts of the PPR is the restoration and management of large semi-permanent or permanent wetlands, also known as shallow lakes (mean depth <1.5 m) [14,15]. Shallow lakes and other wetlands in the PPR are depressional and receive water inputs from precipitation and, in some cases, groundwater, which can lead to variable water levels within and between years [16,17]. Varying water levels lead to cyclical changes in vegetation, with the germination of emergent vegetation and annuals during drought years and the growth of submergent vegetation in wet years [18,19]. Additionally, variations in basin morphometry and water levels throughout a single wetland may create different degrees of vegetation density [20]. Breeding marsh birds rely on the emergent vegetation for shelter and building nests [21–23], while aquatic vegetation can provide forage and habitat for prey items [24,25]. However, >50% of wetlands were drained due to agriculture and human settlement during the past 200 years [26,27], resulting in a loss of habitat for breeding marsh birds. Additionally, due to their large size, many remaining shallow lakes were never drained and degraded over time as a result of less hydrological variability [28], increased sediment loads [29–31], and chemical and nutrient inputs [32]. Additionally, the invasion by planktivorous fishes increase bioturbation and decrease zooplankton, and collectively these factors increase phytoplankton biomass and decrease emergent and submersed vegetation growth [33–35]. Ultimately, this highly turbid environment provides poor breeding habitat conditions for breeding marsh birds [14,36].

Efforts to restore shallow lakes tend to involve managing water levels to mimic the natural wet-dry cycle [12], along with removing planktivorous fish and nutrients [37–39], in order to improve water and vegetation conditions for wildlife. Both the success of restorations and response of birds after restoration vary. Restoration success is usually based upon the long-term stability and diversity of submergent and emergent vegetation, along with the persistence of a clear-water state [40,41]. For example, Søngergaard et al. (2007) found that while there were several positive short-term effects from restoration efforts (e.g., fish removal, nutrient input reduction) throughout lakes in Europe, the majority of sites returned to a turbid state with little vegetation after 5–10 years. This could be due to a number of factors, including the return of planktivorous fish [38,42] and the release of phosphorous from the sediment [30]. Indeed, several species of birds respond positively within the first few years of restoration and species richness may increase with older restorations [43,44]. However, other studies have found that restored wetlands may not be functionally equivalent to reference wetlands (i.e., not recently degraded) and will have comparatively fewer species or waterbird use [45,46]. In the Midwest, for instance, restored shallow lakes and wetlands may be susceptible to invasion by aggressive or exotic plants given the appropriate conditions [47–49], leading to a monoculture and a lack of open water. Breeding marsh birds have been shown to have a negative association with such plant species [8]. Monitoring restored wetlands throughout the restoration

process and obtaining estimates of density while accounting for imperfect detection can help assess the potential impact of these restoration efforts on breeding marsh birds.

In Iowa, published studies have estimated the detection probability and density of several marsh bird species across the state [50–52], but none have attempted to obtain estimates particular to restored and pre-restored wetlands. The Shallow Lakes Restoration Project (SLRP) is focused on restoring shallow lakes throughout the Iowa PPR in order to improve conditions for wildlife. Since 2006 the Iowa Department of Natural Resources and Ducks Unlimited, Inc. have restored over 38 shallow lakes in the Iowa PPR through the SLRP. The goal of the SLRP is to improve water quality and the vegetation community to increase the establishment of diverse fish, bird, and invertebrate communities [15]. Since its implementation, these shallow lakes have shown improvements in water quality and vegetation structure [15]. However, as they age some shallow lakes appear to be showing either a decline in water quality and vegetation cover or a dramatic increase in the density of aggressive emergent plants [15,53], indicating a need for further management.

Our goals were to identify factors that influence detection probability of breeding marsh birds, including five species of marsh passerines and two species of secretive marsh birds, in restored and non-restored shallow lakes and to compare detection probabilities among these species. We evaluated the effects of five covariates, including percent cattail, cloud cover, observer, temperature, and wind speed, on detection probability of seven species. We also assessed the influence of call type for secretive marsh birds. We predicted percent cattail, observer, and wind speed would have the greatest influence on detection probability. When comparing estimates of detection probability, we hypothesized that marsh passerines would have higher detection probabilities than secretive marsh birds. Finally, we obtained detection-adjusted estimates of density based on the top models for detection probability and compared density estimates among three restoration states: non-restored, younger, and older.

## Methods

### Study area

The PPR covers about 700,000 km$^2$ in the United States and Canada and is characterized by palustrine wetlands, often known as potholes, and lacustrine wetlands [17,27,54,55]. Palustrine wetlands are typically vegetated, depressional wetlands with relatively shallow water depths, while lacustrine wetlands are deeper (>2.5 m) in some parts of the basin and mostly lacking vegetation [56]. In the U.S., emergent wetlands still cover about 20,000 km$^2$, and in Iowa's Des Moines Lobe region about 800 km$^2$ is emergent wetland area (Fig 1) [27]. This area represents the southernmost extent of the Wisconsin glacial advance, which retreated from Iowa about 14,000 years ago [55,57].

In this study, the term "shallow lake" describes both lacustrine and palustrine wetlands [56] that are large and on average <1.5 m deep. The term "restored" refers to palustrine wetlands that were once severely degraded and subsequently restored by manipulating the hydrology to improve water quality and vegetation. Prior to restoration these shallow lakes could be classified as lacustrine wetlands, as they were mostly lacking vegetation [56]. Restored shallow lakes were passively restored (i.e., no seed additions), and they were drained using an existing outlet or recently installed structures to begin the restoration process. Infrastructure, such as water control structures, channels, pipelines, or pumping systems, were installed into nearly all shallow lakes to manage water levels. Fish exclusion structures were also installed at wetlands to prevent the invasion of rough fish, such as black bullhead (*Ameiurus melas*) and common carp (*Cyprinis carpio*). Once the restoration process began, sites were refilled gradually over (ideally) a 2-year period to allow vegetation to reestablish. Likewise, the term "non-restored" refers

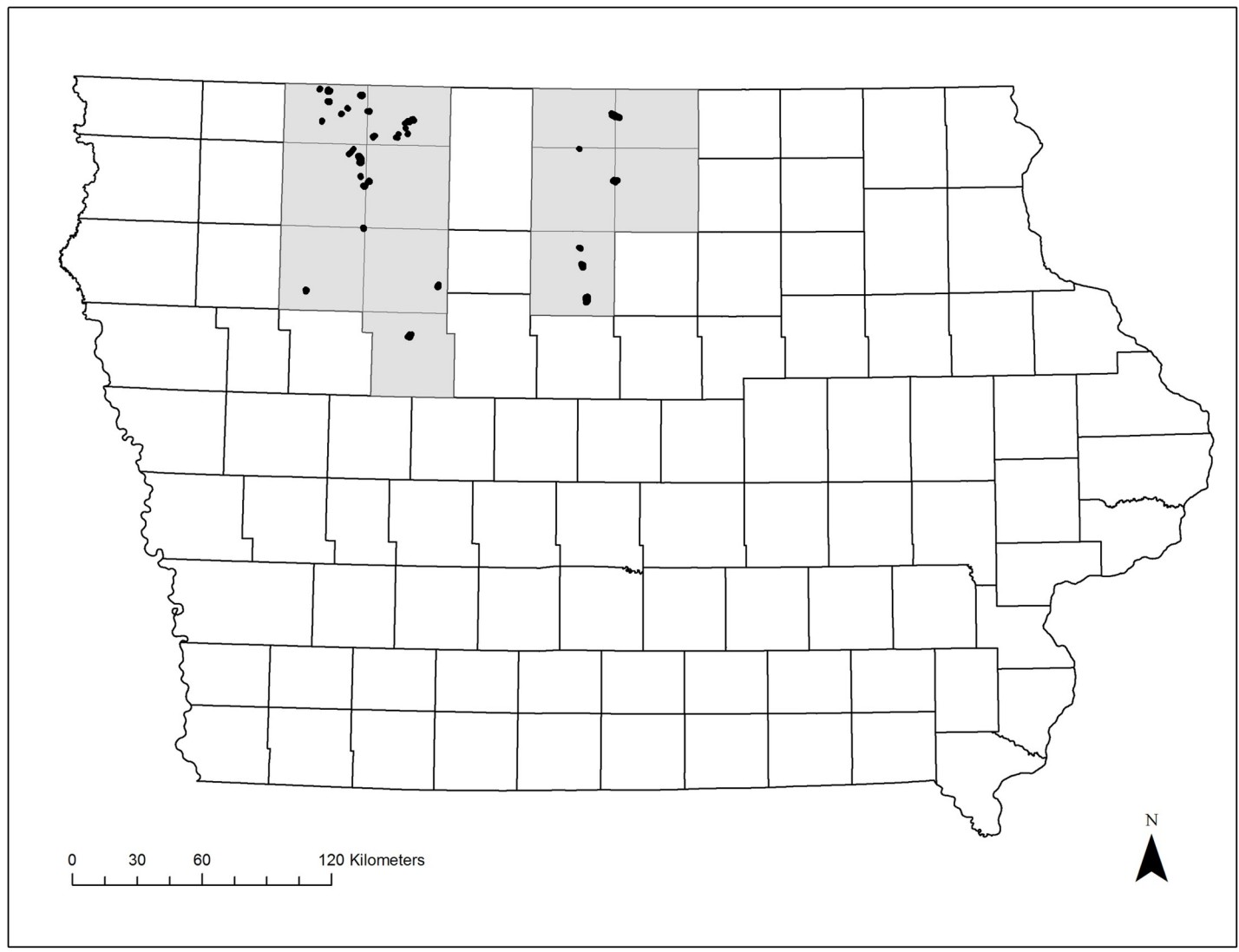

**Fig 1. Location of shallow lakes in the Iowa Prairie Pothole Region (gray outline) for breeding marsh bird surveys conducted in the summer, 2016 and 2017.** Each black dot represents a site, and the 12 shaded counties are those that include the surveyed wetlands.

to lacustrine wetlands that were not manipulated. Most of these shallow lakes were void of emergent vegetation and contained turbid water; some may be restored within the next few years. We considered the date of restoration to be the start of the drawdown, even if it was before completion of the water control structure. Thus, the age of restored shallow lakes ranged from 1 to 11 years.

## Site selection

To examine how the breeding bird communities differed across wetlands in different restoration states, 19 restored sites were randomly chosen based on years since restoration and their large size (>20 ha) spanning the period from 1 to 11 years post-restoration (S1 Appendix). We also chose 11 non-restored shallow lakes to examine pre-restoration bird use of shallow lakes. One of these shallow lakes was in the early stages of restoration in 2017 and, therefore, considered to be a restored site in that year. Sites were located in 12 Iowa counties: Buena Vista,

Calhoun, Cerro Gordo, Clay, Dickinson, Emmet, Hancock, Palo Alto, Pocahontas, Winnebago, Worth, and Wright. A total of 30 shallow lakes were surveyed in the spring of 2016 and 2017. For further analyses, we grouped the shallow lakes into one of three groups (hereafter referred to as restoration states): non-restored, younger (1–5 years since restoration), and older (6–11 years since restoration). Several studies have examined the changes in the vegetation community of restored sites at different ages [19,58,59], and others have found that restorations considered to be "older" may differ biologically from "younger" sites [60,61].

## Bird surveys

We conducted unlimited-radius 10-min point counts throughout each wetland [62]. Points were situated randomly in wetlands, and the number of points depended on the size of the wetland. Two points were placed in wetlands with areas ranging from 10.1–20 ha, three points in sites 20.1–30 ha, four points in sites 20.1–40 ha, and five points in sites >40 ha [63]. Points were situated >400 m apart to avoid double-counting individual birds [7]. We surveyed sets of points twice during each year to account for any seasonal variation in the vocalization rates of species [64,65]. Point counts were conducted between a half-hour before sunrise and up to four hours after sunrise. We did not survey in rainy conditions or when winds exceeded 20 km/h [7]. We recorded several environmental covariates at the start of each survey, and they included temperature (˚C), wind speed (km/h), and cloud cover. We estimated cloud cover using four categories: (0) few or no clouds, (1) partly cloudy, (2) overcast, and (3) fog. We only surveyed in foggy conditions when visibility exceeded 0.25 miles. Upon initial detection, the observer recorded the exact distance (m) to the bird using a laser range-finder (Nikon 8397 Aculon AL11 Laser Rangefinder, Nikon, Melville, New York).

To improve the detection of secretive marsh birds (e.g., rails, bitterns), we incorporated call-broadcasts into the survey period according to methods described by the North American Marsh Bird Monitoring Protocol [6,7]. These species are often difficult to detect using traditional survey techniques [64]. The first 5 min of each survey was a passive period (i.e., no recordings) followed by 5 min of recorded calls. Each minute corresponded to a species; the first 30 s included a recording of the particular species, followed by 30 s of silence. The sequence of calls we used, from first to last, was Least Bittern, Sora (*Porzana carolina*), Virginia Rail (*Rallus limicola*), King Rail (*Rallus elegans*), and Pied-billed Grebe (*Podilymbus podiceps*) [7]. Except for the King Rail, these are regular breeders in Iowa. In previous call-broadcast surveys conducted in Iowa [63], the King Rail did not have many detections, but Sora and Virginia Rail tended to respond to King Rail calls as readily as they responded to intraspecific calls. We used an MP3 player (SanDisk Sansa Clip 1GB, SanDisk Corporation, Milpitas, California) attached to portable speakers (JBL Flip 3, Harman International Industries, Inc., Stamford, Connecticut) and broadcast at 90 dB from a distance of 1 m in front of the speakers [7]. The speaker faced the interior of the wetland and was 0.5 m from the ground or water surface [63]. Our methods for recording detections of secretive marsh birds were the same as for marsh passerines, but we also recorded the call types of detected secretive marsh birds. Secretive marsh birds use a variety of calls throughout the breeding period [66], and there is evidence that detection probability may differ among the different call types [65]. The North American Marsh Bird Monitoring Protocol lists some of the call types given by each species, and we assigned each detection to a call type based on these protocols [6,7].

Because we were not handling birds, we were not required to have approval by the Iowa State University Institutional Animal Care and Use Committee. Our general bird work is authorized under USGS Bird Banding Permit #23285 to Stephen J. Dinsmore, but permits were not needed for this specific study.

## Habitat surveys

After each survey at every point we recorded seven habitat covariates following methods suggested by [6] and similar to [67]. Within 50 m of each point, we visually estimated the percent cover of the dominant vegetation types, which tended to include cattail (*Typha* sp.), bulrush (*Schoenoplectus* sp.), river bulrush (*Bulboschoenus fluviatilis*), or reed canary grass (*Phalaris arundinacea*) to the nearest 5%. We also estimated the percent cover of open water. Other less dominant vegetation types were grouped into a single category and were not used for analysis. Together these values summed to 100%. For analyses, we used an average value for each habitat measurement for the two visits.

## Statistical analysis

To obtain abundance and density estimates of breeding marsh birds, we used the Distance package [68] in Program R [69]. We pooled data across years to increase our sample size and because we had no *a priori* reason to suspect detection probability varied by year. For each species we used the same modeling approach to select the most appropriate model for estimating parameters of interest. To reduce error related to distance sampling we assigned raw distances to distance bins [70]. We did this by visually examining histograms of distances and assigned bins to avoid heaping of distances [70]. We first examined four models recommended by [70]: the hazard-rate model with a cosine expansion, the half-normal model with a Hermite polynomial expansion, the uniform model with a cosine expansion, and the uniform model with a simple polynomial expansion. We fit a null model for each of these functions. For these and all other models, we stratified by restoration state to get estimates of density for non-restored, younger, and older restorations. For determining covariates that influenced detection probability, we used the multiple covariate distance sampling engine (MCDS), and this limited the key functions to the hazard-rate and the half-normal functions [71,72]. We determined the most appropriate key function for this step by comparing Akaike's Information Criterion corrected for small sample sizes ($AIC_c$) [73]. For marsh passerines, we examined the effect of five covariates: observer, percent cattail, cloud cover, temperature, and wind speed. There were three observers throughout the 2-year period. One observer (Observer 1) surveyed wetlands during both years, while the other two observers surveyed in either 2016 or 2017. These two observers received training prior to completing surveys, but they were equally less experienced than Observer 1. Thus, we included the novice observers as a single observer (Observer 2) in the analysis. We assessed quadratic effects of percent cattail, temperature, and wind speed and compared these to the corresponding main effects models. We only included the quadratic effect in the subsequent step if it performed better ($\Delta AIC_c < 2$) than the main effects model. For secretive marsh birds we also assessed the effect of call type. This included four call types for Pied-billed Grebe: (1) *owhoop*, (2) *hyena*, (3) both *owhoop* and *hyena* given within a 10-min period, and (4) other [6,7]. For Virginia Rail this included two categories: (1) *grunt* and (2) other. The other category included less common call types and also visual detections, which were much fewer than aural detections for both species. Finally, we examined all possible combinations of the covariates for each species and used the best competing model to report estimates of density for each restoration state and an overall detection probability. We determined models to be competitive if they were <2 ΔAICc of the top model and they were not more complex versions of the top model [74,75]. For competitive models we assessed beta coefficients of covariates and determined them to have a significant effect on detection probability if the 95% confidence intervals did not overlap zero. If observer or call type were included in competitive models we used the predict function (mrds package) [76] in Program R to produce estimates of detection probability for each category, with all other covariates held

at their means. We used the top model for each species to estimate density for each restoration state. We determined densities to be significantly different among the restoration states if the 95% confidence intervals did not overlap.

## Results

We conducted surveys at 131 points on 30 wetlands during the course of two breeding seasons. Survey effort was similar at non-restored sites (n = 94), younger restorations (n = 86), and older restorations (n = 82). Surveys occurred from 2 June to 15 July in 2016 and from 1 June to 20 July in 2017. We focused our analysis on seven species for which we had an adequate number of detections, and these included Virginia Rail, Pied-billed Grebe, Marsh Wren (*Cistothorus palustris*), Swamp Sparrow (*Melospiza georgiana*), Common Yellowthroat (*Geothlypis trichas*), Red-winged Blackbird (*Agelaius phoeniceus*), and Yellow-headed Blackbird (*Xanthocephalus xanthocephalus*; Table 1). Of the focal species, 1,645 total birds were detected in 2016 and 2,330 total birds in 2017. Red-winged Blackbird had the most detections across all years and surveys (n = 1,000), while Virginia Rail had the fewest detections (n = 140).

The top model for Virginia Rail included the half-normal key function and the effect of percent cattail (Table 2). Percent cattail had a negative effect on detection probability, whereas the effect of temperature was not significant (Table 3). Overall detection probability was 0.12 ± 0.02 SE and density was 0.69 ± 0.15 SE birds/ha. Densities of Virginia Rail were highest in restored shallow lakes, with the greatest densities occurring in older restorations, but this was not a significant difference (Fig 2).

For Pied-billed Grebe, the most competitive model included the half-normal key function with the covariates wind speed, observer, and a quadratic effect of cattail (Table 2). Wind speed had a negative effect on detection probability (Table 3). Observer had a significant effect on detection probability, with a greater detection probability for Observer 1 than for Observer 2. With all other variables held at their means, detection probability was 0.34 and 0.26 for Observer 1 and Observer 2, respectively. Percent cattail had a positive quadratic effect on detection probability but this quadratic component was not significant. Call type and temperature were also included in other competitive models. The most frequently used call was the *owhoop* call (294 detections), followed by the *hyena* (78 detections), other (66 detections), and both the *owhoop* and *hyena* (46 detections). Detection probability was greatest for the *owhoop* call, with a detection probability of 0.36 when all other variables were held constant, and 0.24, 0.27, and 0.28 for the *hyena*, both *hyena* and *owhoop*, and other call types, respectively. In competing models, temperature tended to have a positive effect on detection probability but this was not a significant relationship. Overall detection probability for Pied-billed Grebe was 0.30 ± 0.02 SE and density was 0.44 ± 0.06 SE birds/ha. Densities were greatest in restored shallow lakes, but there was no difference between younger and older restorations (Fig 2).

For Marsh Wrens, the most competitive model included the hazard-rate key function and a quadratic effect of percent cattail and a quadratic effect of wind speed (Table 2). Detection probability mostly declined with percent cattail, but there was as slight increase as percent cattail approached 100% (Table 3). Wind speed showed a negative quadratic effect, with detection probability peaking around 9 kph. Other models within 2 ΔAICc of the top model were more complex versions of the top model so we did not consider them to be competitive. Overall detection probability was 0.15 ± 0.02 SE birds/ha and density was 4.24 ± 0.59 SE birds/ha. Densities of Marsh Wrens were significantly greater in restored sites than in non-restored sites (Fig 2).

Due to a limited number of detections for Swamp Sparrows, we only assessed models with no more than three covariates for this species to avoid overfitting. The single most competitive model for this species included the hazard-rate key function and a linear effect of percent

**Table 1. Total number of detections and percent of points with detections of breeding marsh birds surveyed at shallow lakes in the Prairie Pothole Region of Iowa, summer 2016 and 2017.**

| Species | Year | Restoration State | Total detections | Percent of points with detections |
|---|---|---|---|---|
| Virginia Rail | 2016 | Non-restored | 12 | 15 |
| | | Older | 27 | 46 |
| | | Younger | 36 | 27 |
| | 2017 | Non-restored | 5 | 6 |
| | | Older | 44 | 33 |
| | | Younger | 16 | 23 |
| Pied-billed Grebe | 2016 | Non-restored | 15 | 13 |
| | | Older | 67 | 61 |
| | | Younger | 159 | 73 |
| | 2017 | Non-restored | 13 | 13 |
| | | Older | 179 | 69 |
| | | Younger | 92 | 80 |
| Marsh Wren | 2016 | Non-restored | 60 | 34 |
| | | Older | 135 | 89 |
| | | Younger | 214 | 93 |
| | 2017 | Non-restored | 91 | 43 |
| | | Older | 262 | 85 |
| | | Younger | 138 | 87 |
| Swamp Sparrow | 2016 | Non-restored | 13 | 15 |
| | | Older | 31 | 43 |
| | | Younger | 34 | 27 |
| | 2017 | Non-restored | 44 | 36 |
| | | Older | 77 | 57 |
| | | Younger | 38 | 40 |
| Common Yellowthroat | 2016 | Non-restored | 19 | 26 |
| | | Older | 26 | 46 |
| | | Younger | 56 | 50 |
| | 2017 | Non-restored | 73 | 68 |
| | | Older | 121 | 81 |
| | | Younger | 58 | 70 |
| Red-winged Blackbird | 2016 | Non-restored | 90 | 57 |
| | | Older | 103 | 89 |
| | | Younger | 210 | 86 |
| | 2017 | Non-restored | 198 | 79 |
| | | Older | 287 | 89 |
| | | Younger | 112 | 90 |
| Yellow-headed Blackbird | 2016 | Non-restored | 30 | 23 |
| | | Older | 112 | 79 |
| | | Younger | 196 | 70 |
| | 2017 | Non-restored | 45 | 28 |
| | | Older | 311 | 91 |
| | | Younger | 126 | 93 |

cattail (Table 2), which had a positive effect on detection probability (Table 2). Overall detection probability was 0.12 ± 0.08 SE, while overall density was 0.93 ± 0.62 SE birds/ha. There was no a significant difference among densities in the restoration states (Fig 2).

**Table 2. Model selection results for detection probability for seven species of breeding marsh birds surveyed at shallow lakes in the Prairie Pothole Region of Iowa, summer 2016 and 2017.**

| Species | Model | AICc | ΔAICc | K |
|---|---|---|---|---|
| Virginia Rail | percent cattail | 112.16 | 0.00 | 4 |
| Pied-billed Grebe | percent cattail + percent cattail$^2$ + wind speed + observer | 920.85 | 0.00 | 5 |
| | percent cattail + percent cattail$^2$ + wind speed | 921.13 | 0.28 | 4 |
| | percent cattail + percent cattail$^2$ | 921.52 | 0.67 | 3 |
| | call type + percent cattail + percent cattail$^2$ + wind speed | 922.64 | 1.79 | 5 |
| | percent cattail + percent cattail$^2$ + temperature + wind speed | 922.49 | 1.64 | 5 |
| | observer + percent cattail + percent cattail$^2$ | 922.47 | 1.62 | 4 |
| Marsh Wren | percent cattail + percent cattail$^2$ + wind speed + wind speed$^2$ | 2256.58 | 0.00 | 5 |
| Swamp Sparrow | percent cattail | 521.40 | 0.00 | |
| Common Yellowthroat | temperature + cloud cover + wind speed | 1004.14 | 0.00 | 4 |
| | percent cattail + cloud cover + wind speed | 1005.29 | 1.15 | 4 |
| | wind speed + cloud cover | 1005.60 | 1.45 | 3 |
| Red-winged Blackbird | percent cattail + percent cattail$^2$ + temperature + wind speed | 3232.62 | 0.00 | 5 |
| | percent cattail + percent cattail$^2$ + temperature + cloud cover | 3232.80 | 0.19 | 5 |
| | percent cattail + percent cattail$^2$ + temperature | 3232.80 | 0.18 | 4 |
| | percent cattail + percent cattail$^2$ | 3233.40 | 0.78 | 3 |
| | percent cattail + percent cattail$^2$ + wind speed | 3233.47 | 0.86 | 4 |
| | percent cattail + percent cattail$^2$ + cloud cover | 3234.17 | 1.55 | 4 |
| | percent cattail + percent cattail$^2$ + cloud cover + wind speed | 3234.48 | 1.86 | 5 |
| Yellow-headed Blackbird | percent cattail + percent cattail$^2$ + temperature | 2245.31 | 0.00 | 5 |
| | percent cattail + percent cattail$^2$ + wind speed | 2246.03 | 0.72 | 5 |
| | percent cattail + percent cattail$^2$ | 2246.77 | 1.46 | 3 |

The most competitive model for Common Yellowthroat included the hazard-rate key function along with the covariates temperature, cloud cover, and wind speed (Table 2). Temperature and wind speed did not show significant effects, whereas cloud cover had a negative effect on detection probability (Table 3). Percent cattail was also included in a competing model but did not show a significant effect on detection probability. Overall detection probability was 0.25 ± 0.05 SE and density was 0.40 ± 0.09 SE birds/ha. There was no significant difference among densities in the restoration states (Fig 2)

**Table 3. Beta coefficients (with standard errors) on the logit scale included in the most competitive models for breeding marsh birds surveyed at shallow lakes in the Prairie Pothole Region of Iowa, summer 2016 and 2017.**

| Covariate | Virginia Rail | Pied-billed Grebe | Marsh Wren | Swamp Sparrow | Common Yellowthroat | Red-winged Blackbird | Yellow-headed Blackbird |
|---|---|---|---|---|---|---|---|
| Cloud cover | | | | | *-0.35 (0.13) | | |
| Percent cattail | *-0.14 (0.05) | *-0.26 (0.09) | *-0.55 (0.09) | *0.48 (0.36) | | *-0.21 (0.08) | *-0.70 (0.10) |
| Percent cattail$^2$ | | 0.12 (0.07) | *0.18 (0.07) | | | *0.31 (0.07) | *0.28 (0.09) |
| Temperature | | | | | 0.18 (0.1) | 0.11 (0.06) | 0.14 (0.07) |
| Temperature$^2$ | | | | | | | 0.07 (0.05) |
| Wind speed | | *-0.09 (0.06) | *0.20 (0.07) | | 0.26 (0.13) | 0.10 (0.06) | |
| Wind speed$^2$ | | | *-0.05 (0.04) | | | | |

Significant ($P<0.05$) beta coefficients are indicated with an asterisk (*).

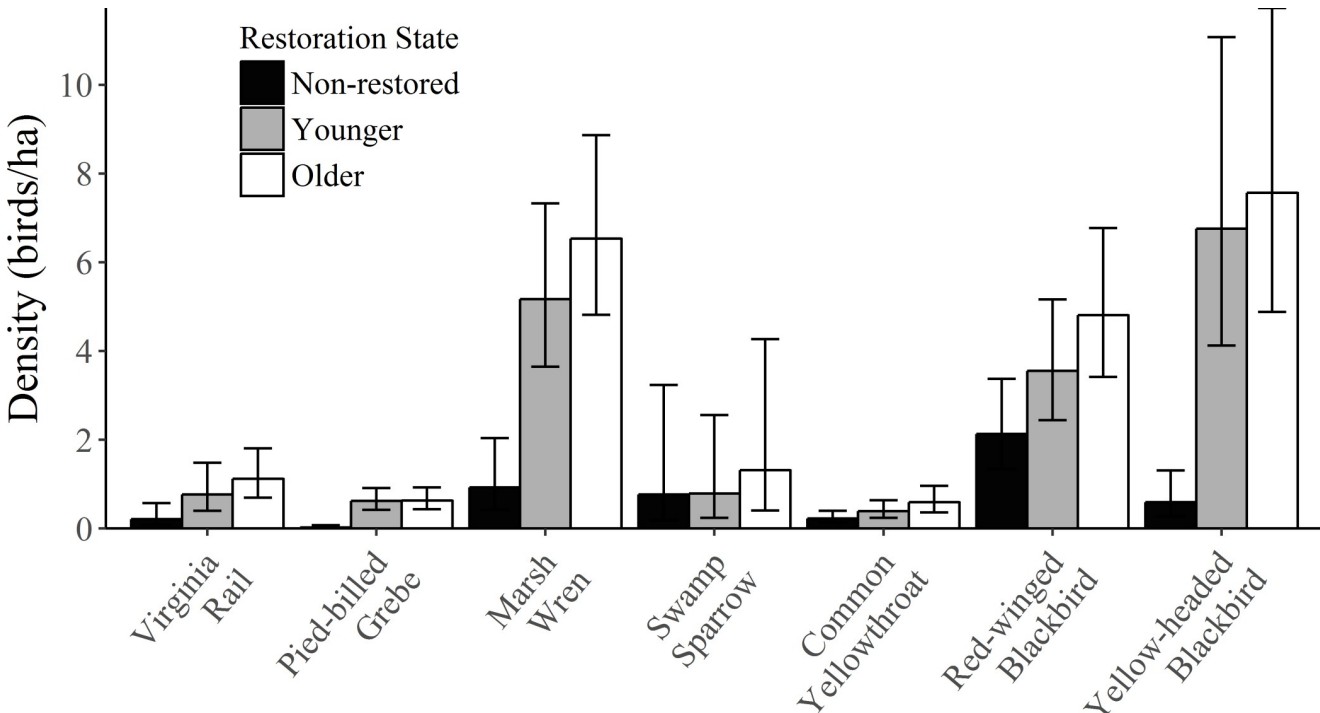

**Fig 2. Densities (birds/ha) of breeding marsh birds across shallow lakes surveyed in the Prairie Pothole Region of Iowa, summer 2016 and 2017.** Sites are divided into non-restored, younger (1–5 years since restoration), and older (6–11 years since restoration) shallow lakes. Vertical error bars represent 95% confidence intervals.

The top model for Red-winged Blackbird included the hazard-rate key function and a quadratic effect of percent cattail, along with temperature and wind speed (Table 2). Percent cattail showed a positive quadratic effect, with detection probability highest at 100% cover and lowest at about 50% cover (Table 3). Temperature and wind speed did not show significant effects on detection probability. Cloud cover was also included in two other competitive models and showed a negative effect with detection probability. Overall detection probability was 0.07 ± 0.009 SE and density was 3.46 ± 0.53 SE birds/ha. Density tended to be greater in older restorations, but this was not significantly greater than younger restorations or non-restored sites (Fig 2).

The most competitive model for detection of Yellow-headed Blackbird included a model with the hazard-rate key function and quadratic effects of percent cattail and temperature (Table 2). Percent cattail showed a positive quadratic effect, with detection probability decreasing sharply between 0 and 60% cattail cover and increasing slightly up to 100% cattail cover (Table 3). Temperature did not have a significant effect on detection. All competitive models included the quadratic effect of percent cattail, and a negative quadratic effect of wind speed was also included in a competing model. Overall detection probability for this species was 0.10 ± 0.02 SE. Densities of Yellow-headed Blackbird were highest in older restorations and lowest in younger restorations (Fig 2), with an overall density of 5.07 ± 1.00 SE birds/ha.

## Discussion

Our findings emphasize the importance of accounting for factors that may influence detection probability when estimating density or abundance. Furthermore, this is one of the few studies that has examined breeding birds at restored, large, shallow lakes in the PPR (but see [37,77–

79]). At least four variables significantly influenced the detection probability of breeding birds in these restorations, which will be important to consider when completing future monitoring efforts [1]. And as with other studies [44,80,81], restoration seemed to improve breeding bird densities when compared to non-restored sites, specifically for species such as Pied-billed Grebe, Marsh Wren, and Yellow-headed Blackbird.

Detection probability varies among avian species, and this is particularly true for breeding marsh birds [52,82,83]. We expected cryptic species such as the Virginia Rail to have the lowest detection probabilities, while more vociferous species such as Yellow-headed Blackbirds and Marsh Wrens would have higher detection probabilities. Indeed, the detection probability of Virginia Rail was relatively low but similar to estimates found in another study in this region [51]. We estimated a higher detection probability for Pied-billed Grebe. This species appears to be more aggressive than Virginia Rails during the breeding period and tended to vocalize more frequently, so this pattern was not surprising. The marsh passerines tended to have higher detection probabilities, with Common Yellowthroat having the highest detection probability at 0.40. While this species was not as abundant as the Yellow-headed Blackbird or Marsh Wren, other studies have shown that Common Yellowthroats generally have a high probability of detection and availability and found similar estimates of this parameter [84–86]. In contrast, we found lower detection probabilities for Yellow-headed Blackbirds and Marsh Wrens, two species that prefer deep wetlands with dense vegetation [67]. While we did not record call types of these species, these species also make a variety of calls, and they can change as the breeding season progresses [52]. Yellow-headed Blackbirds, for example, tend to sing less once young have fledged, so during this phase the probability of detecting this species given it was present likely declined [52].

Previous work has examined the detection probability of marsh birds and how several factors influence detection of different species [23,50,52,66], and our study further highlights variables that should be considered when surveying for breeding marsh birds. Of particular concern is the experience of observers [87,88]. Providing extensive training in identifying species and determining distance from observer are critical to obtaining robust estimates of detection and abundance [7,89,90]. We found that observer influenced the detection probability of Pied-billed Grebes, with the more experienced observer (Observer 1) having a higher detection probability than the less experienced observers (Observer 2). While the less experienced observers received training on identifying secretive marsh birds, the variety of calls of these species appears to make them less likely to be detected by novice observers. However, call type did not appear in the best supported model for either Virginia Rail or Pied-billed Grebe, but it was included in a competitive model for Pied-billed Grebe. Few studies have examined the effects of call type on detection probability of breeding marsh birds (but see [65]), yet there is evidence that detectability varies among call types of secretive marsh birds [6,65]. The *owhoop* call, which tends to be used during courtship and territorial displays, had a higher detectability than the *hyena* call, which is used more for greeting, or other calls [6]. It could be that the *owhoop* call is more easily detected because it is elicited during more aggressive displays and given more frequently than the other call types.

We expected environmental and habitat variables to influence detection probability, but we found this to be true for only a few species. Wind speed, for example, tends to negatively influence detectability of some species, as windier conditions make hearing birds more difficult [51,91]. However, others have found no effect of wind speed on detectability or availability [92], and some have found a positive influence [93]. We found a negative trend for Pied-billed Grebe, but wind speed had a negative quadratic effect on the detectability of Marsh Wrens, with detectability peaking around 9 kph. While temperature did not show a significant effect for any species, this covariate tended to have a positive relationship with detection probability

for the Common Yellowthroat, Red-winged Blackbird, and Yellow-headed Blackbird. Temperature has been shown to have a slight positive effect on some breeding marsh birds, as it may increase the activity of birds [94], but others have found little to no effect [51,95]. On the other hand, cloud cover showed a negative influence the detection of Common Yellowthroats. Other studies have found an influence of cloud cover on detection probability [51,64], but the reason is still unclear. Furthermore, while covariates related to weather are important, habitat may also influence the behavior of birds and affect their detection probability [96]. We found that the percent cover of cattail had a negative influence on the detection probability of Virginia Rails. While this species prefers nesting sites with dense, tall vegetation cover, such conditions can make detecting them more difficult due decreased visibility and noise coming from moving vegetation. On the other hand, percent cattail had a positive effect on detection probability for Swamp Sparrows and a positive quadratic effect on detection probability for Marsh Wren, Red-winged Blackbird, and Yellow-headed Blackbird. Although vegetation can decrease visibility, it also provides more perching locations as birds sing, which could further increase detection probability.

Accounting for the various covariates that influence detection probability allowed us to obtain reliable estimates of density at shallow lakes in three different restoration states when detection probability was less than one. For all species density tended to be greater in older restorations than in younger restorations, but we did not find significant differences between these two restoration states for any species. For three species (Pied-billed Grebe, Marsh Wren, and Yellow-headed Blackbird) densities were significantly greater in both older and younger restorations than in non-restored sites. Yellow-headed Blackbirds and Marsh Wrens, obligate wetland breeders, build nests using robust vegetation [16,97–99]. Pied-billed Grebes also prefer to nest in wetlands with emergent vegetation and prefer some degree of interspersion between emergent vegetation and open water [21,22,67]. The diversity of emergent and submergent vegetation, along with the development of sedge meadow and wet prairie habitat near the edges, may provide habitat for a more diverse community of invertebrates and seeds, which make up significant portions of the diets for several species [24,97,98,100]. While we did not directly measure the presence of sedge meadow and wet prairie vegetation, as most of our points were located away from shallow lake edges, there did appear to be a greater presence and abundance of these habitats at restored sites. On the other hand, we did not find significant differences in density for Common Yellowthroat, Swamp Sparrow, and Red-winged Blackbird. These species are not considered to be obligate wetland breeders and will nest in other habitat types, which could explain this finding [101–103].

Detection probability is an important parameter to consider when estimating the density of birds at wetland restorations. Several of these species can be difficult to detect given the training required to conduct surveys and the habitat that may influence an observer's ability to survey affectively, and other environmental factors can also influence detection probability. Future monitoring should focus on minimizing the effects of environmental variables by standardizing protocols and on providing training to surveyors, particularly for identifying secretive marsh birds. We did not assess the influence of habitat variables on density or abundance, yet our findings showed that restored shallow lakes are home to increased densities of several breeding marsh birds.

## Supporting information

**S1 Appendix. Names, counties, and UTM coordinates of shallow lakes in the Iowa Prairie Pothole Region for breeding marsh birds, summer 2016 and 2017.** Restored shallow lakes are indicated by an asterisk.
(XLSX)

## Acknowledgments

Funding for this project was provided by the U.S. Fish and Wildlife Service (Prairie Pothole Joint Venture, grant number F15AP00612) and the Iowa Department of Natural Resources (Resource Enhancement and Protection License Plate Funds). We thank S. Moodie and J. Mize for assistance with field work. We thank B. J. Wilsey and T. W. Stewart for providing edits and advice on field methods and analyses. We also thank M. Gulick, T. Bishop, D. Janke, B. Hellyer, S. Woodruff, T. J. Herrick, C. Maddix, C. LaRue, K. Kinkead, K. Murphy, and T. Harms for help with planning and logistics.

## Author Contributions

**Conceptualization:** Rachel A. Vanausdall, Stephen J. Dinsmore.

**Data curation:** Rachel A. Vanausdall.

**Formal analysis:** Rachel A. Vanausdall, Stephen J. Dinsmore.

**Funding acquisition:** Stephen J. Dinsmore.

**Investigation:** Rachel A. Vanausdall.

**Methodology:** Rachel A. Vanausdall, Stephen J. Dinsmore.

**Supervision:** Stephen J. Dinsmore.

**Visualization:** Rachel A. Vanausdall.

**Writing – original draft:** Rachel A. Vanausdall.

**Writing – review & editing:** Stephen J. Dinsmore.

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
