## [Decision Letter · Decision Letter 0]

28 Oct 2019

PONE-D-19-26684

Detection and density of breeding marsh birds in Iowa wetlands

PLOS ONE

Dear Vanausdall,

Thank you for submitting your manuscript to PLOS ONE. After careful consideration, we feel that it has merit but does not fully meet PLOS ONE’s publication criteria as it currently stands. Therefore, we invite you to submit a revised version of the manuscript that addresses the points raised during the review process.

Your paper should be acceptable subject to the revisions raised by the reviewer.  I am sorry that I could not find further reviewers to look at your paper.

We would appreciate receiving your revised manuscript by Dec 12 2019 11:59PM. To enhance the reproducibility of your results, we recommend that if applicable you deposit your laboratory protocols in protocols.io, where a protocol can be assigned its own identifier (DOI) such that it can be cited independently in the future. For instructions see: http://journals.plos.org/plosone/s/submission-guidelines#loc-laboratory-protocols

We look forward to receiving your revised manuscript.

Kind regards,

Maura (Gee) Geraldine Chapman, PhD DSc

Academic Editor

PLOS ONE

Journal Requirements:

1. 

2.  In your Methods section, please provide additional location information of the study sites, including geographic coordinates for the data set if available.

Reviewers' comments:

Reviewer's Responses to Questions

**Comments to the Author**

1. Is the manuscript technically sound, and do the data support the conclusions?

Reviewer #1: Yes

2. Has the statistical analysis been performed appropriately and rigorously? 

Reviewer #1: Yes

3. Have the authors made all data underlying the findings in their manuscript fully available?

Reviewer #1: Yes

4. Is the manuscript presented in an intelligible fashion and written in standard English?

Reviewer #1: Yes

5. Review Comments to the Author

Reviewer #1: I thought the paper was well written, informative work that could have some impact for future studies.

The methods section:

Line 110 - you may want to define what palustrine / lacustrine systems are - mentioned several times but I didn't see a definition for the terms. Is it a difference in water regime, wetland size, water depth? Veg community different?

What are the native vegetation communities for the wetlands you studied - what is the goal for restoration? You mentioned restoration processes for the small lakes (removing fish, removing water structures, mimicking natural water regimes) and allowing a 2 year period for vegetation to reestablish - is the type of vegetation important? Could this have had some impact on detection for secretive marsh birds (eg - least bittern have a different habitat preference than sora or viginia rail)? The changes in vegetation community is mentioned several times, but there isn't really any explanation as to what that community is. A brief explanation of what a successful restoration looks like could be helpful here.

Was distance to roads taken into account when you established your survey points, or was the effect of disturbance from traffic negligible?

At line 402 you mention the development of sedge meadow and wet prairie habitat near the edges of the shallow lakes as a positive for the bird communities - are these common for restored shallow lakes in Iowa?

Overall - good paper, I think the issues I found were very minor, and it could be published without too many changes.

6. PLOS authors have the option to publish the peer review history of their article (what does this mean?). If published, this will include your full peer review and any attached files.

Reviewer #1: Yes: Eric D Wilson

---

## [Author Response · Author response to Decision Letter 0]

2 Dec 2019

Editor’s comments

2. In your Methods section, please provide additional location information of the study sites, including geographic coordinates for the data set if available.

Response: We included a list of the 30 shallow lakes, along with the county and UTM coordinates. These were included as supplementary material (S1 Table).

Response: Technically, we are not required to have any permits for this work because we were not handling birds. Our general bird work is authorized under USGS Bird Banding Permit #23285 to Stephen J. Dinsmore. We included a statement at the end of the “Bird surveys” section that states: “Because we were not handling birds, we were not required to have approval by the Iowa State University Institutional Animal Care and Use Committee.”

Response: We will deposit our data in Iowa State University’s DataShare if the publication is accepted. 

 

Reviewer 1 Comments

I thought the paper was well written, informative work that could have some impact for future studies.

Response: We appreciate this statement.

The methods section:

Line 110 - you may want to define what palustrine / lacustrine systems are - mentioned several times but I didn't see a definition for the terms. Is it a difference in water regime, wetland size, water depth? Veg community different?

Response: Thank you for these comments and questions. We defined the terms palustrine and lacustrine in the Study Area section. Based on Cowardin et al. (1979), one of the biggest differences between these is the presence or absence of vegetation. Lacustrine wetlands generally lack vegetation, while palustrine wetlands are characterized by hydrophytes.

What are the native vegetation communities for the wetlands you studied - what is the goal for restoration? You mentioned restoration processes for the small lakes (removing fish, removing water structures, mimicking natural water regimes) and allowing a 2 year period for vegetation to reestablish - is the type of vegetation important? Could this have had some impact on detection for secretive marsh birds (eg - least bittern have a different habitat preference than sora or viginia rail)? The changes in vegetation community is mentioned several times, but there isn't really any explanation as to what that community is. A brief explanation of what a successful restoration looks like could be helpful here.

Response: We included more explanation on the vegetation community and the meaning of restoration success in the introduction. Specifically, we further described vegetation characteristics of prairie shallow lakes and how they can be impacted by hydrological fluctuations, invasive fish species, and excessive nutrient loading. We also emphasized the meaning of a successful restoration based on work by other researchers. 

Was distance to roads taken into account when you established your survey points, or was the effect of disturbance from traffic negligible?

Response: We did not take into account distance to roads. We believe disturbance from roads was negligible because most of the roads near these shallow lakes were either (a) dead-end access roads to the wetland, or (b) rural roads that were not heavily used. 

At line 402 you mention the development of sedge meadow and wet prairie habitat near the edges of the shallow lakes as a positive for the bird communities - are these common for restored shallow lakes in Iowa?

Response: The presence of sedge meadow and wet prairie habitat was not quantified, as most of our surveys were completed away from shallow lake edges. However, the authors and observers did notice a greater abundance and presence of such habitat while completing these surveys. We clarified this in the discussion. 

Overall - good paper, I think the issues I found were very minor, and it could be published without too many changes.

---

## [Editor Report · Decision Letter 1]

31 Dec 2019

Detection and density of breeding marsh birds in Iowa wetlands

PONE-D-19-26684R1

Dear Dr. Vanausdall,

We are pleased to inform you that your manuscript has been judged scientifically suitable for publication and will be formally accepted for publication once it complies with all outstanding technical requirements.

With kind regards,

Maura (Gee) Geraldine Chapman, PhD DSc

Academic Editor

PLOS ONE
---

## [Editor Report · Acceptance letter]

15 Jan 2020

PONE-D-19-26684R1 

Detection and density of breeding marsh birds in Iowa wetlands 

Dear Dr. Vanausdall:

I am pleased to inform you that your manuscript has been deemed suitable for publication in PLOS ONE. Congratulations! Your manuscript is now with our production department. 

With kind regards,

on behalf of

Professor Maura (Gee) Geraldine Chapman 

Academic Editor

PLOS ONE